# Epigenetic age-predictor for mice based on three CpG sites

Yang Han[1,2], Monika Eipel[1,2], Julia Franzen[1,2], Vadim Sakk[3],
Bertien Dethmers-Ausema[4], Laura Yndriago[5], Ander Izeta[5,6], Gerald de Haan[4],
Hartmut Geiger[3,7], Wolfgang Wagner[1,2]*

[1]Helmholtz-Institute for Biomedical Engineering, Stem Cell Biology and Cellular Engineering, RWTH Aachen University Medical School, Aachen, Germany; [2]Institute for Biomedical Engineering – Cell Biology, University Hospital RWTH Aachen, Aachen, Germany; [3]Institute of Molecular Medicine, Ulm University, Ulm, Germany; [4]Laboratory of Ageing Biology and Stem Cells, European Research Institute for the Biology of Ageing, University Medical Center Groningen, Groningen, Netherlands; [5]Tissue Engineering Laboratory, Instituto Biodonostia, San Sebastian, Spain; [6]Department of Biomedical Engineering, School of Engineering, Tecnun-University of Navarra, San Sebastian, Spain; [7]Experimental Hematology and Cancer Biology, Cincinnati Children's Hospital Burnet Campus, Cincinnati, United States

**Abstract** Epigenetic clocks for mice were generated based on deep-sequencing analysis of the methylome. Here, we demonstrate that site-specific analysis of DNA methylation levels by pyrosequencing at only three CG dinucleotides (CpGs) in the genes *Prima1*, *Hsf4*, and *Kcns1* facilitates precise estimation of chronological age in murine blood samples, too. DBA/2 mice revealed accelerated epigenetic aging as compared to C57BL6 mice, which is in line with their shorter life-expectancy. The three-CpG-predictor provides a simple and cost-effective biomarker to determine biological age in large intervention studies with mice.
DOI: https://doi.org/10.7554/eLife.37462.001

*For correspondence:
wwagner@ukaachen.de

## Introduction

Age-associated DNA methylation (DNAm) was first described for humans after Illumina Bead Chip microarray data became available to enable cross comparison of thousands of CpG loci (*Bocklandt et al., 2011*; *Koch and Wagner, 2011*). Many of these age-associated CpGs were then integrated into epigenetic age-predictors (*Hannum et al., 2013*; *Horvath, 2013*; *Weidner et al., 2014*). However, site-specific DNAm analysis at individual CpGs can also provide robust biomarkers for aging. For example, we have described that DNAm analysis at only three CpGs enables age-predictions for human blood samples with a mean absolute deviation (MAD) from chronological age of less than five years (*Weidner et al., 2014*). Such simplistic age-predictors for human specimen are widely used because they enable fast and cost-effective analysis in large cohorts.

Recently, epigenetic clocks were also published for mice by using either reduced representation bisulfite sequencing (RRBS) or whole genome bisulfite sequencing (WGBS) (*Petkovich et al., 2017*; *Stubbs et al., 2017*; *Wang et al., 2017*). For example, Petkovich et al. described a 90 CpG model for blood (*Petkovich et al., 2017*), and Stubbs and coworkers a 329 CpG model for various different tissues (*Stubbs et al., 2017*). Nutrition and genetic background seem to affect the epigenetic age of mice – and thereby possibly aging of the organism (*Cole et al., 2017*; *Hahn et al., 2017*; *Maegawa et al., 2017*). In analogy, epigenetic aging of humans is associated with life expectancy, indicating that it rather reflects biological age than chronological age (*Lin et al., 2016*; *Marioni et al.,*

**eLife digest** Epigenetic marks are chemical modifications found throughout the genome – the DNA within cells. By influencing the activity of nearby genes, the marks govern developmental processes and help cells to adapt to changes in their surroundings. Some epigenetic marks can be gained or lost with age. A lot of aging research focuses on one type of mark, called "DNA methylation". By measuring the presence or absence of specific methyl groups, scientists can estimate biological age – which may differ from calendar age.

Recent studies have developed computer models called epigenetic aging clocks to predict the biological age of mouse cells. These clocks use epigenetic data collected from the entire genomes of mice, and are useful for understanding how the aging process is affected by genetic parameters, diet, or other environmental factors. Yet, the genome sequencing methods used to construct most existing epigenetic clocks are expensive, labor-intensive, and cannot be easily applied to large groups of mice.

Han et al. have developed a new way to predict biological aging in mice that needs methylation information from just three particular sections of the genome. Even though this approach is much faster and less expensive than other epigenetic approaches to measuring aging, it has a similar level of accuracy to existing models. Han et al. use the new method to show that cells from different strains of laboratory mice age at different rates. Furthermore, in a strain that has a shorter life expectancy, aging seems to be accelerated.

The new approach developed by Han et al. will make it easier to study how aging in mice is affected by different interventions. Further studies will also be needed to better understand how epigenetic marks relate to biological aging.

DOI: https://doi.org/10.7554/eLife.37462.002

*2015*). However, DNAm profiling by deep sequencing technology is technically still challenging, relatively expensive, and not every sequencing-run covers all relevant CpG sites with enough reading depth.

## Results

Therefore, we established pyrosequencing assays for nine genomic regions of previously published predictors (*Petkovich et al., 2017*; *Stubbs et al., 2017*). These regions were preselected to have multiple age-associated CpGs in close vicinity. DNAm was then analyzed in 24 blood samples of female C57BL/6 mice that covered a broad range of 12 different age groups (11 to 117 weeks old). The nine amplicons covered a total of 71 CpG sites (*Supplementary file 1*) and we used machine learning to identify the best fitted model for epigenetic age-predictions using cross-fold validation on the training set. The best results were observed for 15 CpGs from five different amplicons that provided an extremely high correlation with chronological age in the training set ($R^2$ = 0.99; mean absolute deviation [MAD]=2.76 weeks; *Supplementary file 2*), albeit the training set might be too small for this approach. To make the method more easily applicable and more cost-effective, we wanted to focus on less CpGs. When we varied the regularization parameters for models with less CpGs, the precision declined significantly. For example the best model with three CpGs comprised the three CpGs of *Hsf4* (CpGs# 3,4,5) that also revealed the overall highest Pearson correlations with chronological age ($R^2$ = 0.95; MAD = 5.24 weeks). However, combination of different hypo- and hypermethylated amplicons might be advantageous to facilitate better assessment of plausibility of the results. Therefore, we alternatively selected those three CpGs that revealed the highest Pearson correlation with chronological age in different amplicons. These three CpGs were associated with the genes Proline rich membrane anchor 1 (*Prima1*: chr12:103214639; $R^2$ = 0.71), Heat shock transcription factor 4 (*Hsf4*: chr8:105271000; $R^2$ = 0.95) and Potassium voltage-gated channel modifier subfamily S member 1 (*Kcns1*: chr2:164168110; $R^2$ = 0.83; *Figure 1A–C*; *Figure 1—figure supplement 1*). Notably, all three CpGs were derived from the epigenetic age-predictor for blood samples (*Petkovich et al., 2017*). A multivariable model for age-predictions was established for DNAm at the CpGs in *Prima 1* (α), *Hsf4* (β), and *Kcns1* (γ):

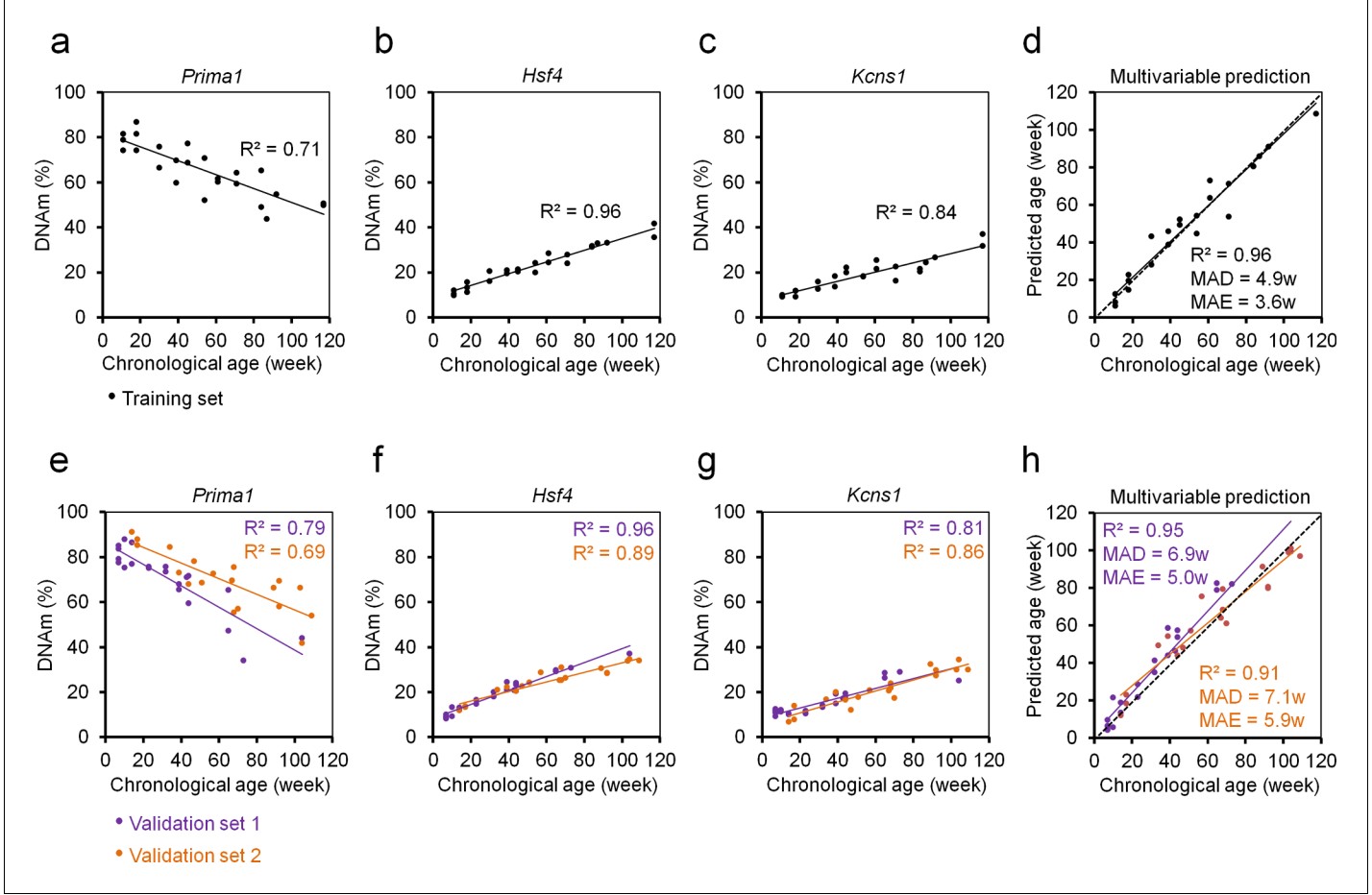

**Figure 1.** Three CpG epigenetic age-predictor for mice. (**a–c**) DNA methylation (DNAm) of three CpGs in the genes *Prima1*, *Hsf4* and *Kcns1* was analyzed by pyrosequencing in 24 C57BL/6 mice (training set). Coefficient of determination ($R^2$) of DNAm *versus* chronological age is indicated. (**d**) Based on these age-associated DNAm changes a multivariable model for age prediction was calculated. (**e–g**) Subsequently, two independent validation sets were analyzed: 21 C57BL/6 mice from the University of Ulm and 19 C57BL/6 mice from the University of Groningen (validation sets 1 and 2, respectively). (**h**) Age predictions with the three-CpG-model revealed a high correlation with chronological age in the independent validation sets (MAD = mean absolute deviation; MAE = median absolute error).

DOI: https://doi.org/10.7554/eLife.37462.003

The following figure supplement is available for figure 1:

**Figure supplement 1.** Target sequences of pyrosequencing assays.

DOI: https://doi.org/10.7554/eLife.37462.004

Predicted age$^{C57BL/6}$ (in weeks) = $-58.076 + 0.25788\ \alpha + 3.06845\ \beta + 1.00879\ \gamma$

Age-predictions correlated very well with the chronological age of C57BL/6 mice in the training set ($R^2$ = 0.96; MAD = 4.86 weeks; *Figure 1D*).

Our three CpG age-predictor was subsequently validated in a blinded manner for 21 C57BL/6J mice (7 to 104 weeks old) from the University of Ulm (validation set 1) and 19 C57BL/6J mice (14 to 109 weeks old) from the University of Groningen (validation set 2). The results of both validation sets revealed high correlations with chronological age ($R^2$ = 0.95 and 0.91, respectively; *Figure 1E–H*) with relatively small MADs (6.9 and 7.1 weeks) and median absolute errors (MAE; 5.0 and 5.9 weeks). Thus, our age-predictions seem to have similar precision as previously described for multi-CpG predictors based on RRBS or WGBS data (*Petkovich et al., 2017*; *Stubbs et al., 2017*; *Wang et al., 2017*).

Gender did not have significant impact on our epigenetic age-predictions for mice (*Figure 2*), as described before (*Maegawa et al., 2017*; *Petkovich et al., 2017*; *Stubbs et al., 2017*). In contrast, the human epigenetic clock is clearly accelerated in male donors (*Hannum et al., 2013*;

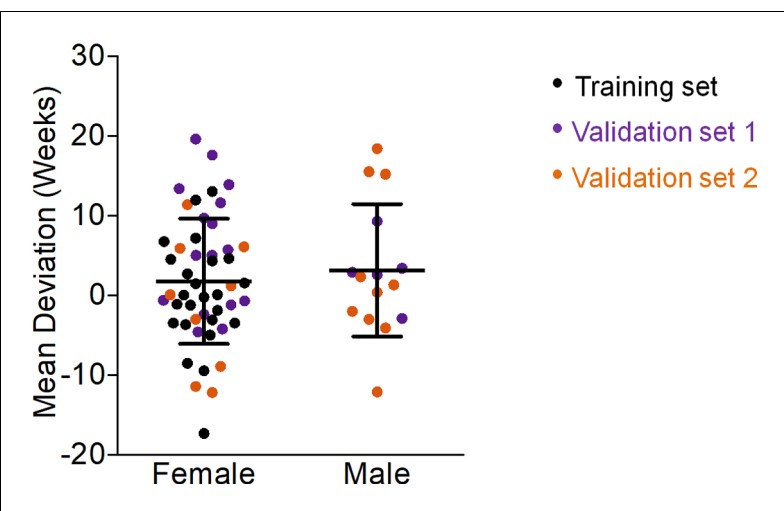

**Figure 2.** Gender does not affect epigenetic age predictions in mice. The deviations of predicted age by our three-CpG predictor *versus* chronological age did not reveal significant differences between female and male C57BL/6 mice (Mann–Whitney U test p=0.6).
DOI: https://doi.org/10.7554/eLife.37462.005

*Horvath, 2013*; *Weidner et al., 2014*). This coincides with shorter life expectancy in men than woman, whereas in mice there are no consistent sex differences in longevity (*Goodrick, 1975*).

To address the question if our three CpG signature was also applicable for other tissues than blood we analyzed the DNAm in skin, kidney, intestine, lung, liver, heart, brain, testis, and pancreas of 3 young (9.6 weeks old) and three old mice (56.9 weeks old). In all tissues tested the samples of old mice were predicted to be older using our three CpG signature. However, the different DNAm levels clearly demonstrate that the model needs to be retrained to be applied for these tissues (*Figure 3*).

Subsequently, we analyzed epigenetic aging of DBA/2 mice that have a shorter life expectancy than C57BL/6 mice (*Goodrick, 1975*) (33 mice from Ulm and Groningen; 6 to 109 weeks old). The three CpGs in *Prima1*, *Hsf4* and *Kcns1* revealed high correlation with chronological age ($R^2$ = 0.91, 0.88 and 0.83, respectively), albeit the offset in DNAm between DBA/2 and C57BL/6 mice indicated that the signature needs to be retrained for different mouse strains (*Figure 4a–c*). Notably, the slopes were higher in DBA/2 mice, particularly for the CpG in *Prima1*. Furthermore, DNAm *of Hsf4* increased at a higher rate in young DBA/2 mice, indicating that it is more accurately modelled as a function of logarithmic age. This has also been described in human for many age-associated CpGs in pediatric cohorts (*Alisch et al., 2012*). In fact, epigenetic age-predictions in DBA/2 mice seemed to follow a logarithmic model of age ($R^2$ = 0.89; *Figure 4d*) rather than a linear association ($R^2$ = 0.86). These results provided evidence for accelerated epigenetic aging of DBA/2 mice.

Either way, epigenetic age-predictions were overall significantly overestimated in the shorter-lived DBA/2 mice, suggesting that age-predictors need to be adjusted for different inbred mice strains. To this end, we have retrained a multivariate model for DBA/2 mice:

Predicted age$^{DBA/2}$ (in weeks) = 87.54294–1.22221 α + 0.991558 β + 0.355444 γ

This adjusted model facilitated relatively precise age-predictions for DBA/2 mice ($R^2$ = 0.95; MAD = 7.1 weeks; MAE = 5.3 weeks; *Figure 4e*).

## Discussion

Generation of confined epigenetic signatures is always a tradeoff between integrating more CpGs for higher precision and higher costs for analysis (*Wagner, 2017*). It was somewhat unexpected that with only three CpGs our signature facilitated similar precision of epigenetic age-predictions as the previously published signatures based on more than 90 CpGs. This can be attributed to the higher precision of DNAm measurements at individual CpGs by bisulfite pyrosequencing, which is one of

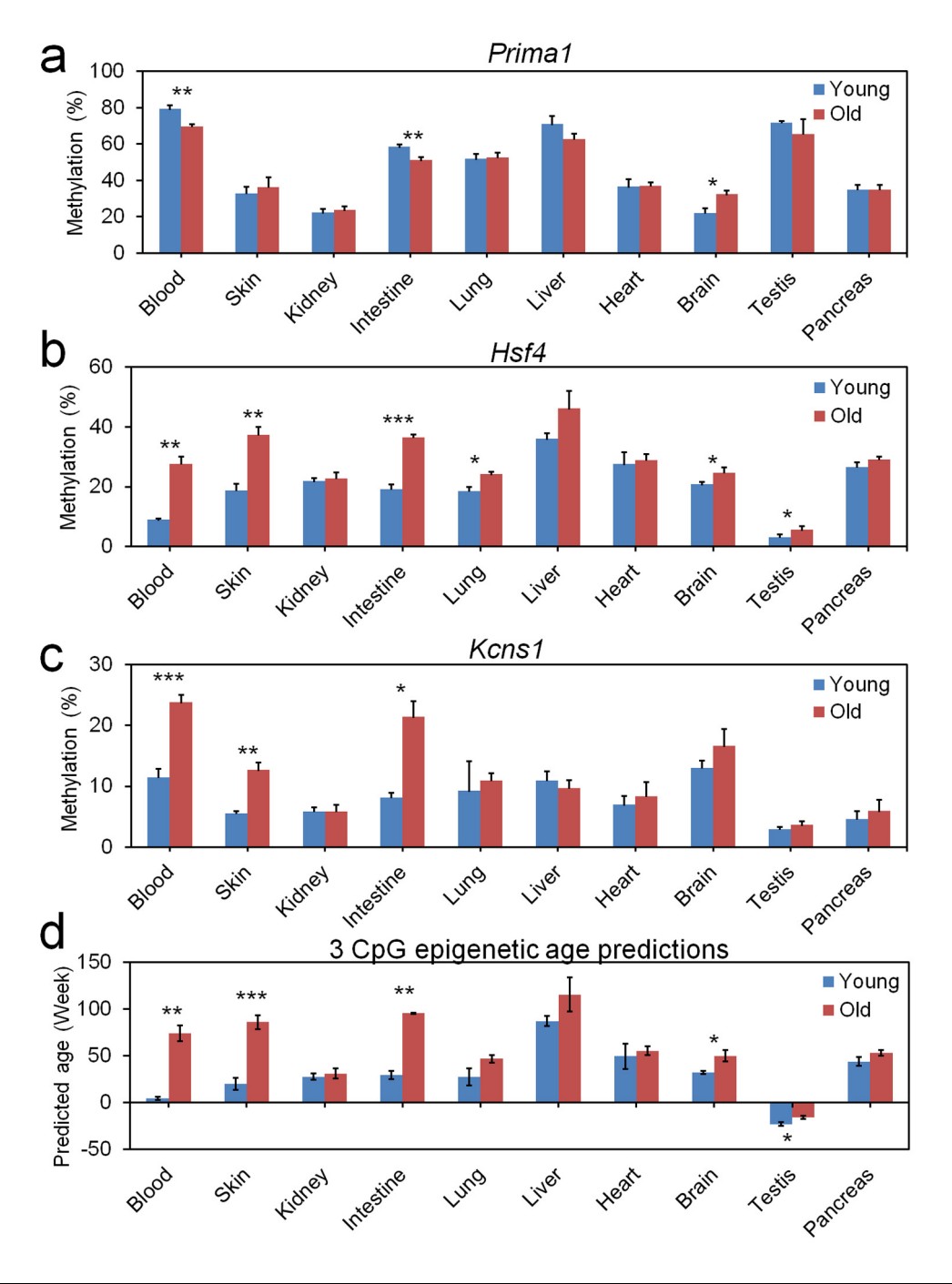

**Figure 3.** Age-associated DNA methylation at the three CpG sites in different tissues. Different tissues were isolated of three young (9.6 weeks) and three old mice (56.9 weeks) and DNAm was analyzed at the three relevant CpGs in (a) *Prima1*, (b) *Hsf4*, and (c) *Kcns1*. Epigenetic age-predictions using the 3 CpG model for blood demonstrated also significant differences between young and old mice in skin, intestine, brain, and testis (mean ± standard deviation; Student t-tests: *p<0.05; **p<0.01; ***p<0.001).
DOI: https://doi.org/10.7554/eLife.37462.006

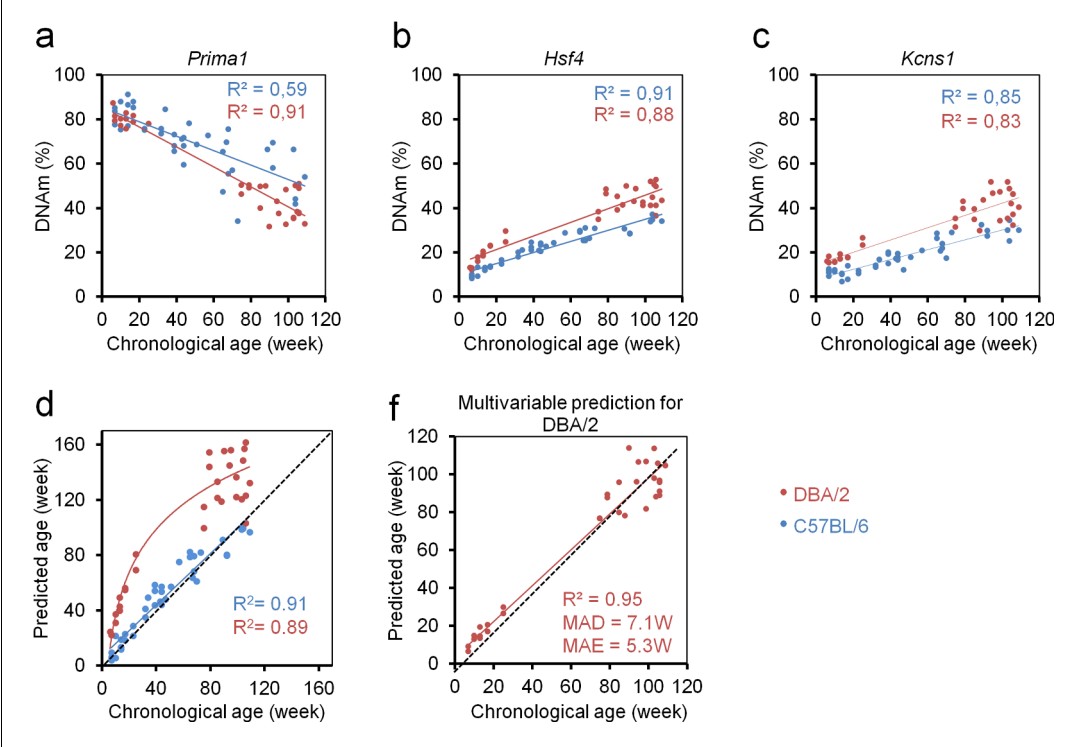

**Figure 4.** Epigenetic aging is accelerated in DBA/2 mice as compared to C57BL/6 mice. (**a–c**) Age-related DNA methylation (DNAm) determined by pyrosequencing assay for three candidate CpGs on 33 of DBA/2 blood samples (14 mice from the University of Ulm and 19 mice from the University of Groningen; red). For comparison we provided measurements of the C57BL/6 mice (only from validation sets; blue). (**d**) Epigenetic age-predictions using the three CpG multivariable model for the C57BL/6 mice (blue; linear regression) and DBA/2 mice (red, logarithmic regression). Age-predictions in DBA/2 mice rather followed a logarithmic regression (R = Pearson correlation); (**e**) Based on the DNAm measurements in DBA/2 we adjusted the multivariate regression model for age-predictions of this mouse strain as described in the text (DBA/2 predictor).
DOI: https://doi.org/10.7554/eLife.37462.007

the most precise methods for determining DNAm at single CpG resolution (**BLUEPRINT consortium, 2016**). Particularly in RRBS data not all CpG sites are covered in all samples and a limited number of reads notoriously entails lower precision of DNAm levels at these genomic locations. Thus, genome wide deep sequencing approaches facilitate generation of robust large epigenetic age-predictors, while site specific analysis may compensate by higher precision of DNAm measurement at individual CpGs.

The ultimate goal of epigenetic age-predictors for mice is not to develop near perfect age predictors, but to provide a surrogate for biological aging that facilitates assessment of interventions on aging. In fact, using deep sequencing approaches (RRBS or WGBS) several groups already indicated that relevant parameters that affect aging of the organism - such as diet, genetic background, and drugs - do also impact on epigenetic aging (**Cole et al., 2017**; **Hahn et al., 2017**; **Maegawa et al., 2017**). It is yet unclear if epigenetic aging signatures can be specifically trained to either correlate with chronological age or biological age. For humans, recent studies indicate that this might be possible (**Levine et al., 2018**) and we have previously demonstrated that even individual age-associated CpGs can be indicative for life expectancy (**Zhang et al., 2017**). Further studies will be necessary to gain better understanding how epigenetic age predictions are related to the real state of biological aging, and how it is related to alternative approaches to quantify biological aging, such as telomere length (**Belsky et al., 2018**).

Our three CpG model has been trained for blood samples – a specimen that is commonly used in biochemical analysis and the small required volume can be taken without sacrificing the mice. However, epigenetic aging may occur at different rates in different tissues. It is difficult to address this question in humans because it is difficult to collect samples of various tissues in large aging cohorts,

whereas this is feasible in mice. We demonstrate that age-associated DNAm changes occur in multiple tissues in our three CpGs albeit they were initially identified in blood (*Petkovich et al., 2017*). Furthermore, DNAm levels may vary between different hematopoietic subsets (*Frobel et al., 2018*; *Houseman et al., 2014*). In the future, sorted subsets should be analyzed to determine how the three CpG signature is affected by blood counts.

The results of our three CpG signature suggest that epigenetic aging is accelerated in DBA/2 mice. Notably, in elderly DBA/2 mice the epigenetic age predictions revealed higher 'errors' from chronological age, which might be attributed to the fact that the variation of lifespan is higher in DBA/2 than C57BL/6 mice (*de Haan et al., 1998*; *Goodrick, 1975*). It will be important to validate the association of the epigenetic age-predictions with biological age by additional correlative studies, including life expectancy in mice.

Taken together, we describe an easily applicable but quite precise approach to determine epigenetic age of mice. We believe that our assay will be instrumental to gain additional insight into mechanisms that regulate age-associated DNAm and for longevity intervention studies in mice.

## Materials and methods

### Mouse strains and blood collection

Blood samples of C57BL/6J mice of the training set and of the validation set one were taken at the University of Ulm by submandibular bleeding (100–200 µl) of living mice or postmortem from the vena cava. C57BL/6J samples of the validation set two were taken at the University of Groningen from the cheek. DBA/2J samples were taken at the University of Ulm (n = 14) and Groningen (n = 19). All mice were accommodated under pathogen-free conditions. Experiments were approved by the Institutional Animal Care of the Ulm University as well as by Regierungspräsidium Tübingen and by the Institutional Animal Care and Use Committee of the University of Groningen (IACUC-RUG), respectively. To analyze age-associated changes in different tissues we used three young (9.6 weeks old) and three old mice (56.9 weeks old) C57BL/6J mice (JaxMice) in accordance with relevant Spanish and European guidelines after approval by the Biodonostia Animal Care Committee. These mice were sacrificed and dissected immediately. 25 mg of tissue (10 mg in the case of spleen) or 200 µl of blood were used for DNA extraction.

### Genomic DNA isolation and bisulfite conversion

Genomic DNA was isolated from 50 µl blood using the QIAamp DNA Mini Kit (Qiagen, Hilden, Germany). Kidney and liver DNA extractions were digested with Ribonuclease A (100 mg/ml, Sigma R4875). DNA concentration was quantified by Nanodrop 2000 Spectrophotometers (Thermo Scientific, Wilmington, USA). 200 ng of genomic DNA was subsequently bisulfite-converted with the EZ DNA Methylation Kit (Zymo Research, Irvine, USA).

### Pyrosequencing

Bisulfite converted DNA was subjected to PCR amplification. Primers were purchased at Metabion and the sequences are provided in *Supplementary file 3*. 20 µg PCR product was immobilized to 5 µl Streptavidin Sepharose High Performance Bead (GE Healthcare, Piscataway, NJ, USA), and then annealed to 1 µl sequencing primer (5 µM) for 2 min at 80°C. Amplicons were sequenced on Pyro-Mark Q96 ID System (Qiagen, Hilden, Germany) and analyzed with PyroMark Q CpG software (Qiagen).

### Alternative approaches to select CpGs for multivariable models

We used a penalized regression model from the R package glmnet on the training dataset to establish a predictor of mouse age based on CpG methylation. The alpha parameter of glmnet was set to 1 (lasso regression) and the lambda parameter was chosen by cross-fold validation of the training dataset (10-fold cross validation). Alternatively, we trained our multivariable model with preselected CpGs based on location in three different amplicons, high Pearson correlation (R) of DNAm with chronological age, and combination of hyper- and hypomethylated sites.

## Statistical analysis

Linear regressions, MAD and MAE were calculated with Excel. Statistical significance of the deviations between predicted and chronological age was estimated by Mann–Whitney U test or Student´s t-test as indicated.

## Acknowledgements

This work was supported by the Else Kröner-Fresenius-Stiftung (2014_A193; to WW), by the German Research Foundation (DFG; WA 1706/8-1 to WW; and GRK 1789 CEMMA, GRK 2254 HEIST and SFBs 1074, 1149 and 1275 to HG), by the German Ministry of Education and Research (BMBF; 01KU1402B to WW; and SyStarR to HG), and by the NIH (R01HL134617 and R01DK104814 to HG). The Groningen samples were obtained from the Mouse Clinic for Cancer and Ageing (http://www.mccanet.nl), which is supported by a grant from the Netherlands Organization for Scientific Research (NWO). The funding bodies were not involved in study design, data analysis, or writing of the manuscript.

## Additional information

### Competing interests

Wolfgang Wagner: cofounder of Cygenia GmbH that can provide service for Epigenetic-Aging-Signatures (http://www.cygenia.com), but the method is fully described in this manuscript. The other authors declare that no competing interests exist.

### Funding

| Funder | Grant reference number | Author |
| --- | --- | --- |
| Else Kröner-Fresenius-Stiftung | 2014_A193 | Wolfgang Wagner |
| Deutsche Forschungsgemeinschaft | WA 1706/8-1 | Wolfgang Wagner |
| Bundesministerium für Bildung und Forschung | 01KU1402B | Wolfgang Wagner |
| NIH Clinical Center | R01HL134617 | Hartmut Geiger |
| Nederlandse Organisatie voor Wetenschappelijk Onderzoek | | Gerald de Haan |
| Deutsche Forschungsgemeinschaft | GRK 1789 CEMMA | Hartmut Geiger |
| Deutsche Forschungsgemeinschaft | GRK 2254 HEIST | Hartmut Geiger |
| Deutsche Forschungsgemeinschaft | SFBs 1074 | Hartmut Geiger |
| Deutsche Forschungsgemeinschaft | SFBs 1149 | Hartmut Geiger |
| Deutsche Forschungsgemeinschaft | SFBs 1275 | Hartmut Geiger |
| NIH Clinical Center | R01DK104814 | Hartmut Geiger |
| Bundesministerium für Bildung und Forschung | SyStarR | Hartmut Geiger |

The funders had no role in study design, data collection and interpretation, or the decision to submit the work for publication.

### Author contributions

Yang Han, Formal analysis, Writing—original draft, Performed pyrosequencing; Monika Eipel, Formal analysis, Performed pyrosequencing; Julia Franzen, Tested alternative aging models; Vadim Sakk,

Bertien Dethmers-Ausema, Laura Yndriago, Resources; Ander Izeta, Gerald de Haan, Resources, Supervision, Project administration; Hartmut Geiger, Conceptualization, Resources, Supervision, Funding acquisition, Project administration; Wolfgang Wagner, Conceptualization, Formal analysis, Supervision, Funding acquisition, Writing—original draft, Project administration

### Author ORCIDs
Gerald de Haan ⓘ http://orcid.org/0000-0001-9706-0138
Hartmut Geiger ⓘ http://orcid.org/0000-0002-5794-5430
Wolfgang Wagner ⓘ https://orcid.org/0000-0002-1971-3217

### Ethics
Animal experimentation: Experiments were approved by the Institutional Animal Care of the Ulm University as well as by Regierungspräsidium Tübingen and by the Institutional Animal Care and Use Committee of the University of Groningen (IACUC-RUG), respectively. To analyze age-associated changes in different tissues we used 3 young (67 days old) and 3 old (398 days old) C57BL/6J mice (JaxMice) in accordance with relevant Spanish and European guidelines after approval by the Biodonostia Animal Care Committee.

### Decision letter and Author response
Decision letter https://doi.org/10.7554/eLife.37462.014
Author response https://doi.org/10.7554/eLife.37462.015

## Additional files

### Supplementary files
• Source data 1. Pyrosequencing raw data of mouse epigenetic aging predictor.
DOI: https://doi.org/10.7554/eLife.37462.008

• Supplementary file 1. Age-associated DNAm in nine genomic regions of the training set.
DOI: https://doi.org/10.7554/eLife.37462.009

• Supplementary file 2. Multivariable model based on 15 CpGs
DOI: https://doi.org/10.7554/eLife.37462.010

• Supplementary file 3. Primers for pyrosequencing.
DOI: https://doi.org/10.7554/eLife.37462.011

• Transparent reporting form
DOI: https://doi.org/10.7554/eLife.37462.012

### Data availability
Raw data of pyrosequencing is provided as supplemental EXCEL table (Source data 1).

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
