## [Decision Letter]

Thank you for submitting your article "Epigenetic Age-Predictor for Mice based on Three CpG Sites" for consideration by *eLife*. Your article has been reviewed by three peer reviewers and the evaluation has been overseen by a Reviewing Editor and Jessica Tyler as the Senior Editor.

The reviewers have discussed the reviews with one another and the Reviewing Editor has drafted this decision to help you prepare a revised submission. We consider this work more appropriate in the category of a Tools and Resources paper rather than as a Research Article.

Summary:

The submitted study presents a new methylation clock for mouse blood based on analysis of only three CpG sites. This seems to be a useful and practical tool, as previous studies produced accurate methylation clocks based on ~100 sites. The training set included 24 blood samples obtained from C57BL/6 mice representing 12 age groups ranging from 11 to 117 week old. Although the number of samples is not large, fair representation of age groups seems to make the clock usable across ages. The authors focused on 9 genomic regions enriched in age-associated sites obtained from previous studies. Three individual sites with the highest correlation with the training set were then selected, and the clock was built based on a simple multivariate linear model. The validation set included 21 mice from the same site (University of Ulm) and 19 mice from a different site (University of Groningen). Precision of the clock was equal to MAE = 3.6 weeks for the training set and MAE = 5 and MAE = 5.9 weeks for validation sets. This is comparable with the available clocks produced by other methods. The new clock was applied to 25 samples of DBA/2 mice having a shorter lifespan. The clock showed a higher age for these short-lived mice compared to C57BL/6 mice of the same chronological age.

Essential revisions:

1) Methods used for site selection are not completely clear. The authors explain their selection by choosing the sites with maximal correlation with age. However, first, it's not clear why exactly three sites were chosen. Second, selection of sites with maximal individual correlation doesn't guarantee that the multivariate model based on these sites would result in highest precision. To make site selection more convincing, you may apply a machine learning approach (linear model with L1 regularization) to the whole set of sites (all sites from 9 genomic regions) and vary the regularization parameter to obtain models with different numbers of sites. Then, these models can be applied to the validation set 1, and precision can be calculated. In this case, you could show how precision changes with the number of sites (the number of remaining sites in the model on the x axis and precision (R^2^ or MAE) on the y axis). This will tell how much precision you lose when proceeding to the model with fewer sites. Based on this plot, you could select the model with the optimal number of sites (minimal number of sites that provides precision, which doesn't significantly increase with the addition of additional sites). And then apply it to the validation set 2 to get the unbiased estimate of precision. This approach could make the analysis much more convincing and also explain the choice of the number of sites.

2) R^2^ is shown for every training and validation set as a metric of quality. However, in the text it is explained as Spearman correlation. This complicates interpretation of the results as usually the ratio of explained variance is denoted by R^2^, which is equal to the square of Pearson correlation, but not to the Spearman correlation. Please, either change the symbol you use (for example, correlation coefficient is usually denoted as ρ), or explain the R^2^ in the text (for example, specify that this is Spearman correlation squared).

3) You didn't specify the number of age groups used for the development of the clock. From the figure, it seems 12 age groups were used. We recommend adding this information to the text as it supports the analysis (12 age groups is a broad range that makes the results more convincing).

4) Comparison of age prediction for C57BL/6 and DBA/2 mice is questionable. DBA/2 samples represent a narrow range of ages, which includes almost no young mice (based on the figure it appears that only 4 samples represent mice <75 weeks old). This reduces quality of the analysis, as nonlinear behavior is often observed in the old ages, which can partly explain the difference between the ages predicted for C57BL/6 and DBA/2 mice. Development of the clock for DBA/2 samples is even more dependent on the age range. Therefore, quality of the clock built for DBA/2 does not look reliable. Additional samples of young DBA/2 mice could improve quality of the findings. Alternatively, this drawback should be clearly noted in the text and text revised accordingly.

5) In the Abstract, you state "DBA/2J mice revealed accelerated epigenetic aging as compared to C57BL6 mice" In fact, Figure 2 appears to show that the DBA/2 mice are about "40 weeks older" at every age – there is barely any age-associated divergence of the predicted aged for DBA/2 and C57BL/6. In other words, it does not seem as if the DBA/2 are aging faster. Rather, they appear to be born older and remain so throughout life. This is perhaps best explained by a need for re-calibrating the clock in different strains of mice. Figure 4E appears to confirm this. So, we agree with the authors conclusion that "age-predictors should be adjusted for different inbred mice strains" but do not agree that "DBA/2J mice revealed accelerated epigenetic aging as compared to C57BL6 mice."

6) You didn't specify if both training and validation sets or only validation sets of C57BL/6 mice were used when the predicted age was compared between this strain and DBA/2. To make the analysis unbiased from the construction of the clock, only validation sets should be used there. Based on the figure, it seems this was indeed the case, but anyway it should be specified in the text as this is important from the methodological point of view.

7) There is far too much emphasis placed on age prediction. Ultimately, the residual or difference between chronological and epigenetic age is of the most interest. The goal is not to develop near perfect age predictors. In humans, the clocks with the strongest age predictions typically do not contribute the most to differential risk of aging-related conditions, which should be the goal. This point can be addressed by revision of the text.

8) Is this new measure specific to blood? Were any experiments done to validate it in any other tissue or in sorted cells? It is possible that changes in methylation may be confounded by blood cell composition, etc. If it is not possible to address it experimentally within the revision timeframe, it should at the very least be discussed as a limitation.

9) Epigenetic measures in mice will only be useful if (a) they track difference in lifespan/healthspan both between and within strains, and (b) they show response to intervention. We feel like the utility of this clock is being over-sold prior to the necessary validations being shown.

10) "These results provide further evidence that age-associated DNAm is generally rather related to biological age […]" is an overstatement. If the DBA/2J mice are not aging epigenetically, there is little data in this manuscript to support the idea that the epigenetic clock is a measure of biological age as opposed to chronological age. The clock that the authors have reported here was calibrated vs. chronological age and appears to function well as such. However, there is no direct evidence that it reports on biological age.

---

## [Author Response]

Essential revisions:1) Methods used for site selection are not completely clear. The authors explain their selection by choosing the sites with maximal correlation with age. However, first, it's not clear why exactly three sites were chosen. Second, selection of sites with maximal individual correlation doesn't guarantee that the multivariate model based on these sites would result in highest precision. To make site selection more convincing, you may apply a machine learning approach (linear model with L1 regularization) to the whole set of sites (all sites from 9 genomic regions) and vary the regularization parameter to obtain models with different numbers of sites. Then, these models can be applied to the validation set 1, and precision can be calculated. In this case, you could show how precision changes with the number of sites (the number of remaining sites in the model on the x axis and precision (R^2^ or MAE) on the y axis). This will tell how much precision you lose when proceeding to the model with fewer sites. Based on this plot, you could select the model with the optimal number of sites (minimal number of sites that provides precision, which doesn't significantly increase with the addition of additional sites). And then apply it to the validation set 2 to get the unbiased estimate of precision. This approach could make the analysis much more convincing and also explain the choice of the number of sites.

We agree that the suggested machine learning approach is very well suited for selection of CpGs in genome wide DNA methylation data. However, for our bisulfite pyrosequencing assays additional points need to be taken into account: not all genomic regions are suitable to design reliable pyrosequencing assays, precision of pyrosequencing declines towards the end of the reads, and the costs for pyrosequencing runs can be significantly reduced by shorter reads in the sequencing step. Following the reviewers’ advice, we have now done the suggested machine learning approach to identify the best set of CpGs from the 71 CpGs that were covered by the nine amplicons (all sites from 9 genomic regions). These experiments were done by Julia Franzen who is now coauthor on the manuscript. In fact, taking 15 CpGs into account the precision was better in cross-validation with data of the training set and this is now described in the text. On the other hand, the training set is relatively small for this approach and the precision should be further validated in the future. In the validation set, we did not do pyrosequencing of all amplicons and we sequenced only a shorter read length to save some of the costs – thus, we cannot validate the precision of this 15 CpG signature in our independent datasets. Either way, using machine learning to derive more focused signatures with three CpGs the approach selected those three CpGs that revealed the highest Pearson correlations of all 71 CpGs – but all of these would be in the same amplicon. We reasoned that combination of hyper- and hypomethylated CpGs and shorter reads would be advantageous. Thus, we feel that our selection was plausible and this is now described in the text.

2) R^2^ is shown for every training and validation set as a metric of quality. However, in the text it is explained as Spearman correlation. This complicates interpretation of the results as usually the ratio of explained variance is denoted by R^2^, which is equal to the square of Pearson correlation, but not to the Spearman correlation. Please, either change the symbol you use (for example, correlation coefficient is usually denoted as ρ), or explain the R^2^ in the text (for example, specify that this is Spearman correlation squared).

Thank you for notifying us on this mistake. We have now corrected this indicating that R is Pearson correlation.

3) You didn't specify the number of age groups used for the development of the clock. From the figure, it seems 12 age groups were used. We recommend adding this information to the text as it supports the analysis (12 age groups is a broad range that makes the results more convincing).

Indeed, the training set comprised mice of 12 different ages, covering a broad range of ages. As suggested, this is now stated in the manuscript.

Results section: “analyzed in 24 blood samples of female C57BL/6 mice that covered a broad range of 12 different age groups (11 to 117 weeks old).”

4) Comparison of age prediction for C57BL/6 and DBA/2 mice is questionable. DBA/2 samples represent a narrow range of ages, which includes almost no young mice (based on the figure it appears that only 4 samples represent mice <75 weeks old). This reduces quality of the analysis, as nonlinear behavior is often observed in the old ages, which can partly explain the difference between the ages predicted for C57BL/6 and DBA/2 mice. Development of the clock for DBA/2 samples is even more dependent on the age range. Therefore, quality of the clock built for DBA/2 does not look reliable. Additional samples of young DBA/2 mice could improve quality of the findings. Alternatively, this drawback should be clearly noted in the text and text revised accordingly.

This point was very well taken. Following the reviewer’s suggestion, we have now analyzed eight additional samples of young DBA/2 mice. Figures and regression models were adjusted accordingly. In fact, the results clearly support the notion that DNAm followed a non-linear behavior particularly in young DBA/2 mice. The additional results are now in line with our previous assumption that DBA/2 mice reveal faster epigenetic aging. Furthermore, correlation and predictions of the DBA/2 model became much better with the additional measurements (MAE improved from 8.3 weeks to 5.3 weeks).

*5) In the Abstract, you state "DBA/2J mice revealed accelerated epigenetic aging as compared to C57BL6 mice" In fact, Figure 2 appears to show that the DBA/2 mice are about "40 weeks older" at every age – there is barely any age-associated divergence of the predicted aged for DBA/2 and C57BL/6. In other words, it does not seem as if the DBA/2 are aging faster. Rather, they appear to be born older and remain so throughout life. This is perhaps best explained by a need for re-calibrating the clock in different strains of mice. Figure* 4E*appears to confirm this. So, we agree with the authors conclusion that "age-predictors should be adjusted for different inbred mice strains" but do not agree that "DBA/2J mice revealed accelerated epigenetic aging as compared to C57BL6 mice."*

Based on our previous data this comment was very well taken. As mentioned above, we have now measured additional young DBA/2 mice and these results support the notion that there is accelerated epigenetic aging in the young DBA/2 mice. It now became evident that the epigenetic age predictions followed rather a logarithmic function (Figure 4D), as previously reported for human pediatric age predictions by Alisch et al., (Genome Research, 2012).

6) You didn't specify if both training and validation sets or only validations sets of C57BL/6 mice were used when the predicted age was compared between this strain and DBA/2. To make the analysis unbiased from the construction of the clock, only validation sets should be used there. Based on the figure, it seems this was indeed the case, but anyway it should be specified in the text as this is important from the methodological point of view.

In fact, we only used the results of the validation sets of C57BL/6 mice for this comparison and this is now clarified in the figure legend of Figure 4 as suggested.

“For comparison we provided measurements of the C57BL/6 mice (only from validation sets; blue).”

7) There is far too much emphasis placed on age prediction. Ultimately, the residual or difference between chronological and epigenetic age is of the most interest. The goal is not to develop near perfect age predictors. In humans, the clocks with the strongest age predictions typically do not contribute the most to differential risk of aging-related conditions, which should be the goal. This point can be addressed by revision of the text.

This point was well taken and we have now better clarified in the discussion that the residual of chronological and predicted age is of the highest relevance to identifying conditions that impact on epigenetic aging.

For example, in the Discussion section: “The ultimate goal of epigenetic age-predictors for mice is not to develop near perfect age predictors, but to provide a surrogate for biological aging that facilitates assessment of interventions on aging. In fact, using deep sequencing approaches (RRBS or WGBS) several groups already indicated that relevant parameters that affect aging of the organism – such as diet, genetic background, and drugs – do also impact on epigenetic aging (Cole et al., 2017; Hahn et al., 2017; Maegawa et al., 2017).”

8) Is this new measure specific to blood? Were any experiments done to validate it in any other tissue or in sorted cells? It is possible that changes in methylation may be confounded by blood cell composition, etc. If it is not possible to address it experimentally within the revision timeframe, it should at the very least be discussed as a limitation.

Following the reviewer’s advice, we have teamed up with Dr. Ander Izeta and Laura Yndriago, who are now coauthors on the manuscript. They provided 60 samples of young and old mice of 10 different tissues. These samples were analyzed with our 3 CpG signature (new Figure 3). Several tissues revealed significant age-associated changes, similar to blood. On the other hand, the DNAm levels varied between different tissues, indicating that the signature needs to be retrained to test applicability for these tissues. According to the reviewers’ advice, we have also discussed that epigenetic age predictions might be influenced by blood counts and that sorted subsets should be analyzed in the future.

Discussion section: “Furthermore, DNAm levels may vary between different hematopoietic subsets (Frobel et al., 2017; Houseman, Molitor and Marsit, 2014). In the future, sorted subsets should be analyzed to determine how the three CpG signature is affected by blood counts.”

9) Epigenetic measures in mice will only be useful if (a) they track difference in lifespan/healthspan both between and within strains, and (b) they show response to intervention. We feel like the utility of this clock is being over-sold prior to the necessary validations being shown.

We agree with this critical comment and indicated that additional studies are required to unequivocally demonstrate association with lifespan. At least, the additional results with young DBA/2 mice now provide evidence that our epigenetic measures track differences in lifespan between strains. Furthermore, we have preliminary results of another study with dietary interventions which indicate that there might be association with life-expectancy, but these results cannot be integrated into this manuscript since this is ongoing work with another group. In the revised manuscript we have much better discussed the relevance of tracking the difference in lifespan/healthspan.

10) "These results provide further evidence that age-associated DNAm is generally rather related to biological age […]" is an overstatement. If the DBA/2J mice are not aging epigenetically, there is little data in this manuscript to support the idea that the epigenetic clock is a measure of biological age as opposed to chronological age. The clock that the authors have reported here was calibrated vs chronological age and appears to function well as such. However, there is no direct evidence that it reports on biological age.

As indicated above our additional results support the notion that DBA/2 mice are aging epigenetically faster. Furthermore, we have added the following passage to refer to this important issue, which is still under debate.

Discussion section: “It is yet unclear if epigenetic aging signatures can be specifically trained to either correlate with chronological age or biological age. For humans, recent studies indicate that this might be possible (Levine et al., 2018) and we have previously demonstrated that even individual age-associated CpGs can be indicative for life expectancy (Zhang et al., 2017). Further studies will be necessary to gain better understanding how epigenetic age predictions are related to the real state of biological aging, and how it is related to alternative approaches to quantify biological aging, such as telomere length (Belsky et al., 2018).”